# Prevalence and Types of Extended-Spectrum β-Lactamase-Producing Bacteria in Retail Seafood

**DOI:** 10.3390/foods12163033

**Published:** 2023-08-12

**Authors:** Ryan Pearce, Beate Conrady, Luca Guardabassi

**Affiliations:** 1Department of Pathobiology and Population Sciences, Royal Veterinary College, London NW1 0TU, UK; rpearce@rvc.ac.uk; 2Department of Veterinary and Animal Sciences, University of Copenhagen, 2600 Copenhagen, Denmark; bcon@sund.ku.dk

**Keywords:** antimicrobial resistance, seafood, extended-spectrum β-lactamases

## Abstract

**Objectives:** To assess prevalence and types of extended-spectrum β-lactamase (ESBL)-producing bacteria in retail seafood. **Methods:** A literature review was completed according to international guidelines for systematic reviews, except for being performed by a single reviewer. Kruskal–Wallis and Dunn tests were used to determine statistical differences between continents or seafood types. **Results:** Among 12,277 hits, 42 publications from 2011 to 2023 were deemed relevant to the review’s objectives. The median prevalence of ESBL-contaminated products was 19.4%. A significantly lower prevalence was observed in Europe (*p* = 0.006) and Africa (*p* = 0.004) compared to Asia. Amongst the 2053 isolates analyzed in the selected studies, 44.8% were ESBL-positive. The predominant type was CTX-M (93.6%), followed by TEM (6.7%) and SHV (5.0%). Only 32.6% and 18.5% of the CTX-M-positive isolates were typed to group and gene level, respectively. While group 1 (60.2%) was prevalent over group 9 (39.8%) among Enterobacterales, the opposite trend was observed in *Vibrio* spp. (60.0% vs. 40.0%). Information at gene level was limited to Enterobacterales, where CTX-M-15 was the most prevalent (79.2%). **Conclusions:** On average, one in five seafood products sold at retail globally is contaminated with ESBL-producing Enterobacterales of clinical relevance. Our findings highlight a potential risk for consumers of raw seafood, especially in Asia.

## 1. Introduction

Antimicrobial resistance (AMR) poses a significant global threat and is a major public health concern. In 2019, bacterial infections caused 4.95 million deaths, with 1.27 million of those deaths being potentially preventable if the causative agent was susceptible to antimicrobials. This places AMR as the 12th leading cause of death, surpassing HIV and malaria [1]. In Europe alone, AMR is estimated to result in 33,000 deaths annually, with associated healthcare and productivity losses amounting to €1.5 billion [2]. Notably, resistance to β-lactam antimicrobials is responsible for over 70% of AMR-related deaths [3].

Extended-spectrum β-lactamases (ESBL) are enzymes produced by Gram-negative bacteria that inactivate β-lactam antimicrobials [4]. These bacterial enzymes are a major cause of resistance to third- and fourth-generation cephalosporins, which are classified as critically important antimicrobials in human medicine by the World Health Organization (WHO) [3]. ESBL-producing Enterobacterales have been classified by the WHO as “priority pathogens” due to their significant threat to human health and the urgent need for new antibiotics [5]. These bacteria are commonly multidrug-resistant, increasing the risk of mortality and prolonged hospital stays [6]. The ESBL-encoding genes belong to three main families: *bla_CTX-M_*, *bla_TEM_*, and *bla_SHV_* [7]. While TEM and SHV types were predominant in the 1980s and 1990s, CTX-M has progressively become the most prevalent ESBL type. CTX-M is further categorized into six phylogenetic groups (CTX-M-1, CTX-M-2, CTX-M-8, CTX-M-9, CTX-M-25, and CTX-M-151) based on amino acid sequence [8]. CTX-M groups 1 (e.g., CTX-M-1 and CTX-M-15) and 9 (e.g., CTX-M-14 and CTX-M-27) are the most commonly described groups, with CTX-M-15 being the most widespread variant among clinical Enterobacterales worldwide [9].

While human-to-human transmission is the primary source of ESBL-producing *Escherichia coli* carriage in high-income countries [10], there is growing concern about the presence of these AMR determinants in food and their potential transmission to humans [11]. Unlike meat from livestock, seafood is often consumed undercooked or raw, facilitating foodborne transmission. In the Netherlands, a large source attribution study indicated seafood as the most significant non-human source for human carriage of *E. coli* producing CTX-M-15 (10%) and CTX-M-27 (10%) [12]. However, the contributions of different non-human sources to this transmission have yet to be quantified in other parts of the world.

There is scattered information in the scientific literature regarding the extent and characteristics of ESBL contamination in seafood products marketed in different geographical regions. The objective of this review was to gather the available information on prevalence and types of ESBL-producing Gram-negative bacteria in retail seafood products. The data extracted from the selected studies were analyzed to detect significant variations between continents, seafood types and bacterial hosts.

## 2. Materials and Methods

### 2.1. Review Protocol and Registration

We searched for studies investigating prevalence and types of ESBL-producing Gram-negative bacteria in a wide range of retail seafood products, including mollusks, crustaceans, and fish. The review adhered to the guidelines outlined in the Cochrane Handbook for Systematic Review and Interventions, with the exception that the quality and acceptance of the included studies were assessed by a single reviewer [13]. The review was conducted based on a study protocol registered in PROSPERO (registration number: CRD42020207039) [14].

### 2.2. Data Source and Literature Search Strategy

A comprehensive search was conducted on five electronic databases, namely PubMed, ScienceDirect, Web of Science, SCOPUS, and CAB direct. CAB direct additionally contained national reports which were also selected for screening based on our inclusion/exclusion criteria. An initial search was performed in August 2022 and was followed by a second search update in July 2023. The search strategy employed a Boolean combination of free text words and MeSH (Medical Subject Headings) terms to identify relevant records [(“Seafood”, “Fish” OR “Shellfish”) AND (“Extended-spectrum β-lactamase” OR “ESBL) AND (“Antimicrobial resistance” OR “AMR”)].

### 2.3. Selection Process and Inclusion/Exclusion Criteria

We uploaded all the identified studies as PDFs to a bibliographic management software (Mendeley version 1.19.8 for macOS), where duplicates were removed automatically. Additional duplicates were identified and removed manually by the first author (R.P.), who also performed abstract and full-text screening in two separate steps. We included research studies and scientific reports that reported the occurrence of ESBL-contaminated seafood products at retail. Only documents written in English were selected, without any limitation on publication dates. We excluded studies that were based on samples that were not collected at retail level (Figure 1).

### 2.4. Data Extraction

Data extraction was conducted by R.P. using a MS Excel (version 16.75.2) template created to capture relevant information such as publication year, sampling year, country where the study was performed, country where the seafood originated from, seafood type, sample size, number of ESBL-positive samples, ESBL-producing bacterial genera/species, and method for ESBL detection. Whenever available, data on the specific ESBL types, such as CTX-M group or variant, were extracted and included in the analysis. For SHV and TEM, data were only extracted if strains were confirmed to exhibit a ESBL phenotype through susceptibility testing with cefpodoxime, cefotaxime, and/or ceftazidime since some enzymes belonging to these ESBL types (i.e., TEM-1, TEM-2, and SHV-1) are not classified as ESBL.

### 2.5. Data Analysis

The data extracted from the selected studies were tested for normal distribution by using Shapiro–Wilk normality test, which indicated a non-normal distribution (W = 0.8, *p* ≤ 0.001). Accordingly, we used the Kruskal–Wallis test followed by a post hoc Dunn’s test, including the Bonferroni method for *p*-values’ adjustment, to determine statistically significant differences between regions (Europe, Asia, and Africa) and between seafood types (mollusks, crustaceans, freshwater fish, and saltwater fish) in terms of prevalence of ESBL contaminated products. The data were visualized as Box-and-Whisker Plots and the results were presented as the median and the 25th and 75th percentiles. All statistical analyses were implemented in R (Version 3.4.1R Foundation for Statistical Computing, Vienna, Austria). Additionally, the prevalence of the types of ESBL (CTX-M groups, CTX-M variants, SHV, and TEM) contaminating the seafood products and carried by bacteria isolated from those seafood products was calculated as the number of ESBL-contaminated seafood samples divided by the number of seafood samples tested.

## 3. Results

### 3.1. Literature Search Results

Among the 12,277 documents retrieved by our literature search, 42 were deemed relevant based on our inclusion and exclusion criteria (Figure 1). These studies comprised four national reports (i.e., DANMAP from 2018 and NethMap from 2017, 2018 and 2019) and 38 scientific articles (Table 1). The countries in which the studies were conducted included Vietnam (n = 8), South Korea (n = 8), India (n = 4), the Netherlands (n = 3), China (n = 2), Egypt (n = 2), Nigeria (n = 2), Thailand (n = 2), Saudi Arabia (n = 1), Germany (n = 1), Algeria (n = 1), Tanzania (n = 1), Cambodia (n = 1), Spain (n = 1), Myanmar (n = 1), Denmark (n = 1) Portugal (n = 1), Tunisia (n = 1), and Brazil (n = 1). Out of the 42 included studies, 22 (52.4%) reported the origin of the seafood items under study and 20 (90.9%) of them tested seafood items which were domestically produced.

The study from Brazil was excluded from the statistical analysis on geographical differences because it was the only study was available from the Americas. Two additional studies were excluded from this analysis because they did not contain quantitative data on ESBL-contaminated products but were used to report the prevalence of ESBL types in bacteria isolated from seafood. Sixteen studies in which the type of seafood product was not defined were excluded from the statistical analysis of ESBL prevalence among seafood types (i.e., mollusks, crustaceans, freshwater fish, and saltwater fish).

A wide array of culture methods and selective media were used for the isolation and identification of ESBL-producing bacteria (Table 2). Most studies (n = 31/42) involved enrichment and subculture onto selective agar and five of them employed commercial ESBL-selective media. The remaining 11 studies employed direct streaking on selective agar (n = 3) or plating sample wash/homogenate without enrichment (n = 3). The remaining studies did not specify the isolation methods (n = 5). Thirty-seven studies detected putative ESBL producers using the double-disk synergy test, while five studies used minimum inhibitory concentrations (MIC) determined by broth microdilution. Thirty-seven studies typed ESBL genes to either a class, group or gene level using PCR methods, whereas six studies employed whole genome sequencing (WGS) for the identification of ESBL-encoding genes.

### 3.2. Prevalence of ESBL-Contaminated Products

Overall, the median prevalence of products being ESBL-contaminated was 19.4 (Q1 (25th percentile) = 0.6% and Q3 (75th percentile) = 33.3%). The highest prevalence of ESBL-contaminated samples was reported in Thailand (88.6%, 95% CI: 78.0–99.1%), followed by India (64.9%, 95% CI: 57.6–72.1%), Algeria (28.6%, 95% CI: 4.9–52.2%), Vietnam (26.9%, 95% CI: 23.7–30.1%), Saudi Arabia (26.4%, 95% CI: 22.1–30.7%), South Korea (21.6%, 95% CI: 19.1–24.1%), Germany (19.4%, 95% CI: 13.3–25.5%), Cambodia (16.7%, 95% CI: 7.2–26.1%), Spain (16.5%, 95% CI: 9.1–23.9%), Egypt (16%, 95% CI: 11.8–20.2%), Tanzania (13.3%, 95% CI: 8.5–18.0%), Myanmar (9.5%, 95% CI: 0.0–22.1%), China (4.9%, 95% CI: 3.2–6.6%), Nigeria (4.2%, 95% CI: 3.4–4.9%), the Netherlands (2.3%, 95% CI: 1.1–3.4%), Tunisia (1.6%, 95% CI: 1.0–2.2%) Portugal (2%, 95% CI: 0.0–3.2%), and Denmark (0.3%, 95% CI: 0.0–1%).

On a regional level, statically significant differences were observed between Asia and Europe (*p* = 0.006) and between Asia and Africa (*p* = 0.004) (Figure 2). In both cases, Asia was associated with a significantly higher prevalence of ESBL-contaminated products compared to the other continents. There was no statistical difference between Africa and Europe (*p* ≥ 0.05). The median prevalence of ESBL-contaminated products was 1.1% in Africa (Q1 = 0.0%; Q3 = 25.3%), 3.3% in Europe (Q1 = 0.8% and Q3 = 12.2%), and 26.7% in Asia (Q1 = 12.5%; Q3 = 50.0%).

No significant differences were observed between different seafood product types (Figure 3). The prevalence was 8.3% in crustaceans (Q1 = 0.0%; Q3 = 19.7%), 9.6% in freshwater fish (Q1 = 3.3%; Q3 = 29.8%), 24.2% in mollusks (Q1 = 15.8%; Q3 = 26.7%), and 33.3% in saltwater fish (Q1 = 0.6%; Q3 = 50.0%).

### 3.3. Distribution of ESBL Types across Bacterial Hosts, Continents, and Seafood Types

The 42 selected studies reported data on the occurrence of ESBL in 2053 Gram-negative bacterial isolates belonging to *E. coli*, *Klebsiella*, *Salmonella*, *Citrobacter*, *Enterobacter*, *Proteus*, *Providencia*, *Shigella*, *Aeromonas*, *Vibrio*, *Serratia*, *Acinetobacter*, *Pseudomonas*, *Edwardsiella*, *Erwinia*, *Hafnia*, and *Yersinia*. Amongst them, 920 isolates (44.8%, 95% CI: 42.7–47%) were reported to harbor ESBL (Table 3). Enterobacterales accounted for the largest bacterial group (n = 1448). Most ESBL-producing isolates within this order belonged to *E. coli* (73.5%, 95% CI: 70.2–76.3%). The highest prevalence of ESBL-positive isolates was found in Enterobacterales (49%, 95% CI: 46.4–51.5%), followed by *Vibrio* spp. (34.2%, 95% CI: 29.4–38.6%) and *Aeromonas* spp. (25.2%, 95% CI 95% CI: 16.9–33.6%). Within Enterobacterales, the proportion of ESBL-positive isolates varied depending on the species (Table 3). For example, it was higher in *Klebsiella oxytoca* (100.0%) and *Klebsiella pneumoniae* (73.5%) than in *E. coli* (59.5%) and *Salmonella enterica* (18.1%).

CTX-M was by far the most prevalent ESBL type in all bacterial genera isolated from retail seafood (Table 3), accounting for 92.1% (95% CI: 90.3–93.8%) of the ESBL-positive isolates. TEM (6.7%, 95% CI: 5.0–8.3%) and SHV (5.0%, 95% CI: 3.6–6.4%) were observed at a lower prevalence. The co-carriage of CTX-M with SHV and TEM was reported in 0.5% (95% CI: 0.0–0.9%) and 0.8% (95% CI: 0.2–1.3%) of ESBL-positive isolates. CTX-M was also the most common ESBL type regardless of the continent of origin (98.5% in Africa, 92.3% in Asia, and 75.6% in Europe) or seafood type (100% in saltwater fish, 99.3% in mollusks, 97.8% in freshwater fish, 93.1% in crustaceans, and 84.8% in undefined seafood products).

Among 847 CTX-M-positive isolates, 44.0% (95% CI: 40.7–47.4%) were typed to group level, 18.5% (95% CI: 15.9–21.2%) to gene level, and 42.6% were not further typed (95% CI: 38.8–46.4%). Among CTX-M-positive Enterobacterales characterized at the group or gene level, the most frequent groups were 1 (41.8%, 95% CI: 35.9–47.6%) and 9 (58.4%, 95% CI: 52.4–64.1%), and the most frequent gene was CTX-M-15 (79.2%, 95% CI: 71.2–84.2%), followed by CTX-M-55 (9.7%, 95% CI: 5.0–14.2%), CTX-M-3 (0.7%, 95% CI: 0.0–1.9%), and CTX-M-130 (0.7%, 95% CI: 0.0–1.9%). CTX-M was the only ESBL type reported in *Vibrio* spp, which displayed a higher frequency of CTX-M group 9 (60.0%, 95% CI: 45.7–74.3%) compared to group 1 (40.0%, 95% CI: 25.7–54.3%). No data at the group or gene level were available for *Aeromonas* spp. (Table 3).

## 4. Discussion

We investigated the prevalence and types of ESBL-producing bacteria in retail seafood products through a systematic analysis of point prevalence studies available in the scientific literature. Most of the studies included in the review (27/42) originated from Asia. This result may reflect the fact that most of the world’s supply of seafood products originates from this continent [56]. Other regions, including some of the major importing countries such as the United States, Japan, and most European countries, are significantly underrepresented in this review. This observation underlines the inappropriateness of generalizing the results of the review.

Our review reveals that the median prevalence of retail seafood products being contaminated with ESBL-producing bacteria is 19.4% (Q1 = 0.6% and Q3 = 33.3%) globally, meaning that approximately one in five products is contaminated. This result suggests a relatively high risk of exposure to ESBL-producing bacteria among seafood consumers, which, according to a recent survey, represents approximately 64% of the EU population [57]. The annual global consumption of seafood products per capita has more than doubled from almost 9 kg in 1961 to over 20.5 kg in 2018 [58]. The risk of foodborne exposure to ESBL-producing bacteria depends on how seafood products are handled and consumed. Obviously, raw and undercooked seafood poses a higher risk to human exposure [59]. Global data on consumption of raw seafood are lacking, but over the last decade there has been a rise in the popularity of raw or undercooked food, including seafood. According to a recent survey in Portugal, the consumption of raw fish amounts to 6.3 kg per person annually [60]. Altogether, these data suggest that the consumption of raw or undercooked seafood may be an important epidemiological route for the spread of ESBL-producing bacteria in the human population. Indeed, a recent source attribution study in the Netherlands identified seafood as the most common non-human source contributing to carriage of ESBL-producing E. coli in the community [12].

The prevalence of ESBL-contaminated seafood products was higher in Asian studies compared to European and African studies. It should be noted that a large proportion of the studies included in the review (n = 20/42) did not provide data on the country of origin of the seafood under study. In the European studies indicating the origin, most products were imported from Asia (Table 1). Thus, it appears that the prevalence of ESBL-contaminated products is higher in Asian seafood products that are marketed locally compared to those that are exported to Europe. This difference could be due to the quality standards imposed by the European Union for imported products. 

It has been estimated that global antimicrobial consumption in aquaculture is higher than consumption in humans and terrestrial food-producing animals, with the Asia–Pacific region representing the largest share (93.8%) of global consumption [61]. However, factors other than antimicrobial consumption in aquaculture may influence the high prevalence of ESBL-contaminated seafood observed in Asian products, since only 25% of all seafood consumed comes from aquaculture, according to a recent EU survey [57]. It should be noted that approximately 77% of ESBL-producing bacteria reported in seafood belonged to Enterobacterales, which have their natural habitat in the intestinal tract and are generally regarded as indicators of fecal contamination. As the community prevalence of ESBL-producing Enterobacterales is significantly higher in Asia than in Europe [62], this result could simply reflect the widespread occurrence of these bacteria in the local population and be the consequence of pre-harvest fecal contamination of the aquatic environment from which seafood originated or post-harvest human contamination during processing [63]. In support of this notion, it has been hypothesized that the ESBL contamination of seafood is likely part of a larger human meta-cycle, whereby ESBL-producing Enterobacterales of human origin are spread in the community via environmental contamination [12]. Another less plausible explanation for the higher ESBL prevalence in Asian seafood products is the exposure to warmer environmental temperatures, which have been recently associated with increased levels of AMR in terrestrial animals, humans, and the marine environment [64].

Most ESBL types found in seafood isolates belong to CTX-M, which is the most pervasive ESBL type in community and clinical settings [65], with CTX-M-15 being the most common and clinically relevant variant [66]. Unfortunately, 12 out of 42 studies (28.6%) included within this review identified ESBL-producing isolates at the class level only (i.e., CTX-M, TEM, or SHV) without defining ESBL determinants at group or gene level. This additional information could only be retrieved from 30 (71.4%) studies, which unanimously reported high isolation rates for CTX-M group 1 and the pandemic variant CTX-M-15. CTX-M group 9 (mainly CTX-M-14 and CTX-M-27) was the second most common group observed in Enterobacterales and the most common group found among Vibrio spp. There is a known variation in the distribution of ESBL genes across different host species. It has recently been shown that the most common ESBL types in livestock (mainly SHV and CTX-M-1) are distinct from those circulating within the human community (mainly CTX-M-15) [67]. Thus, the observed high prevalence of CTX-M-15 is another example of indirect evidence that ESBL-producing Enterobacterales in retail seafood may originate from humans.

This review has limitations which could impart bias to the study conclusions. The review cannot be defined as “systematic” since it was conducted by a single reviewer, whereas at least two reviewers are required for systematic reviews [68]. A main issue when comparing AMR data generated from different studies is the use of different laboratory methods for ESBL detection, including genotypic and phenotypic methods, which limits adequate comparison between studies [69]. Comparison was further limited by the frequent lack of access to raw data, which made meta-analysis impossible. This highlights the importance of applying FAIR principles in future studies to make scientific data “findable, accessible, interoperable and reusable”, thus maximizing their usefulness and enabling the entire research community to benefit from them [70].

## 5. Conclusions

The current literature indicates that on a global scale, approximately one out of five retail seafood products is contaminated with ESBL-producing bacteria, mainly E. coli and other Enterobacterales producing CTX-M. The prevalence of ESBL-contaminated retail seafood products appears to be higher in Asia compared to Europe. Since seafood is often consumed raw or undercooked, this high prevalence of ESBL in retail seafood can be regarded as a risk to human health. Our findings highlight an urgent need for standardized methods of the surveillance of ESBL-producing Enterobacterales in seafood products to enhance understanding of the actual contribution of the seafood industry to the dissemination of ESBL in the human community.

## Figures and Tables

**Figure 1 foods-12-03033-f001:**
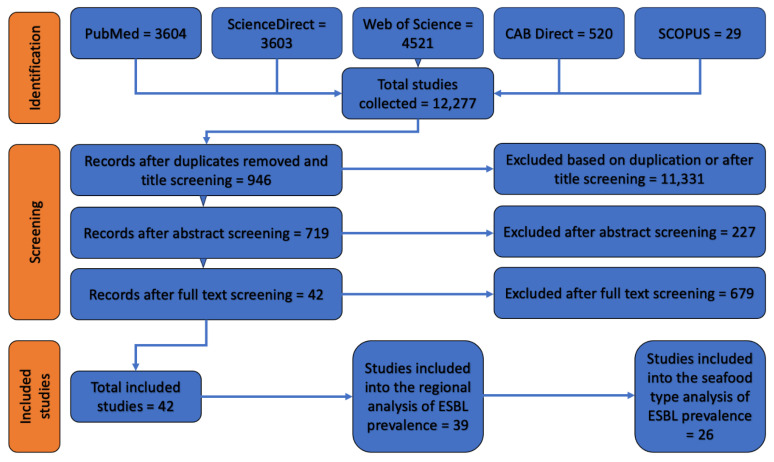
Prisma flow diagram of studies incorporated in the review.

**Figure 2 foods-12-03033-f002:**
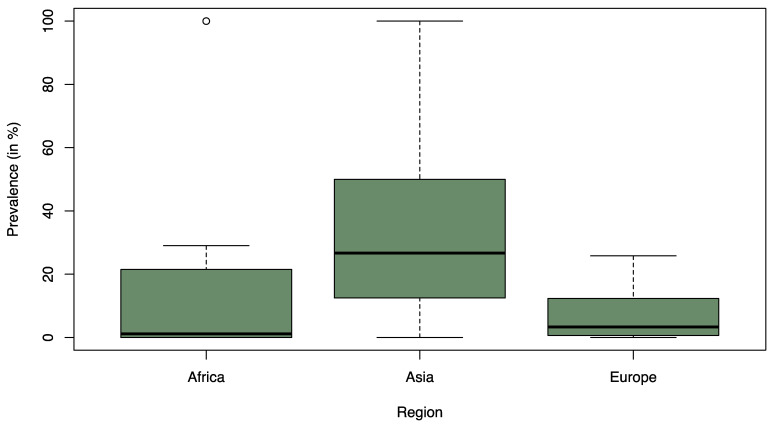
Box-and-Whisker Plot of the recorded ESBL prevalence (%) stratified by region from the 39 studies considered in the statical analysis. The thick line in the Box-and-Whisker Plot represents the median value i.e. 50% of the data are covered. The lower edge of the box represents the 1st quartile i.e. 25% of the data are covered and the upper edge of the box represents the 3rd quartile i.e. 75% of the data are covered. The lowest line outside of the box shows the minimum value and the highest line outside of the box shows the maximum value. The single points outside of the box show the outliers.

**Figure 3 foods-12-03033-f003:**
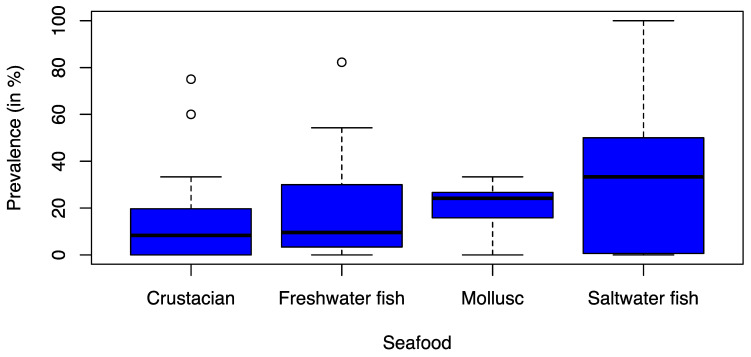
Box-and-Whisker Plot of the recorded ESBL prevalence (%) stratified by different seafood types from the 26 studies considered in the statical analysis. The thick line in the Box-and-Whisker Plot represents the median value i.e. 50% of the data are covered. The lower edge of the box represents the 1st quartile i.e. 25% of the data are covered and the upper edge of the box represents the 3rd quartile i.e. 75% of the data are covered. The lowest line outside of the box shows the minimum value and the highest line outside of the box shows the maximum value. The single points outside of the box show the outliers.

**Table 1 foods-12-03033-t001:** List of selected publications reporting the occurrence of ESBL-producing bacteria in retail seafood products.

Study Type [Reference]	Data Collection/Publication Year	Study Country	Seafood Origin (N Samples)	Seafood Type(s) (N Samples)	Sample Size (N Samples)	ESBL Prevalence (%)
Original article [15]	2018/2020	Korea	Korea (120)	Snail	120	29 (24.2)
Original article [16]	2018/2019	Korea	Korea (145)	Clam	145	36 (24.8)
Original article [17]	2018/2020	Korea	Korea (120)	Snail	120	32 (26.7)
Original article [18]	2013/2015	Vietnam	Vietnam (60)	Shrimp	60	11 (18.3)
Original article [19]	2016/2020	Nigeria	Unknown (1440)	Shrimp	1440	120 (8.3)
Original article [20]	2017/2019	Myanmar	Unknown (21)	Clam	5	1 (20.0)
				Shrimp	16	1 (6.3)
Original article [21]	2014/2018	Thailand	Unknown (35)	Undefined seafood	35	31 (88.6)
Original article [22]	2016/2020	India	Unknown (50)	Sardine	5	1 (20.0)
				Perch	5	1 (20.0)
				Croaker	3	1 (33.3)
				Sea Bass	2	1 (50.0)
				Sea Bream	2	1 (50.0)
				Moon fish	2	1 (50.0)
				Anchovy	3	1 (33.3)
				Lizard fish	4	1 (25.0)
				Belt fish	2	1 (50.0)
				Moon tail	2	1 (50.0)
				Mackerel	2	1 (50.0)
				Gizzard	3	1 (33.3.0)
				Pomfret	2	1 (50.0)
				Shrimp	5	1 (20.0)
				Shrimp	3	1 (33.3.0)
				Clam	3	1 (33.3.0)
				Squid	2	0
Original article [23]	2015/2018	Germany	Bangladesh (14), Denmark (14), Ecuador (12), France (7), Germany (8), India (10), Ireland (12), Italy (17), Netherlands (12), Spain (1), Vietnam (4), Unknown (49)	Shrimp, mussel, clam, cockle	160	31 (19.4)
Original article [24]	2013/2017	India	Unknown (19)	Undefined seafood	19	19 (100.0)
Original article [25]	2018/2019	Korea	Korea (275)	Mussel	275	32 (11.6)
Original article [26]	2016/2019	Nigeria	Unknown (1440)	Shrimp	1440	0
Original article [27]	2012/2016	Saudi Arabia	Thailand (260), India (75), Vietnam (35), Myanmar (35)	Catfish	65	32 (49.2)
				Tilapia	60	18 (30.0)
				Carfoo	50	2 (4.0)
				Tilapia	75	28 (37.3)
				Mirgal	45	3 (6.7)
				Rohu	40	5 (12.5)
				Milkfish	35	19 (54.3)
				Rohu	35	0
Original article [28]	2018/2019	Korea	Korea (120)	Cockle	120	32 (26.7)
Original article [29]	2018/2019	Korea	Korea (120)	Cockle	120	32 (26.7)
Original article [30]	2012/2016	Vietnam	Vietnam (154)	Undefined seafood	101	1 (1.0)
				Shrimp	53	0
Surveillance report [31]	2017/2018	Netherlands	Asia (56)	Undefined seafood	56	7 (12.5)
Surveillance report [32]	2018/2019	Netherlands	Asia (304)	Undefined seafood	304	5 (1.7)
Surveillance report [33]	2019/2020	Netherlands	Asia (304)	Undefined seafood	304	3 (1.0)
Surveillance report [34]	2017/2018	Denmark	Asia (300)	Undefined seafood	300	1 (0.3)
Original article [35]	2019/2020	Brazil	Brazil (1)	Catfish	1	1 (100)
Original article [36]	2016/2018	Cambodia	Unknown (60)	Undefined seafood	60	10 (16.7)
Original article [37]	2018/2020	China	China (200)	Seafood	200	27 (13.5)
Original article [38]	2015/2016	Vietnam	Vietnam (82)	Seafood	82	24 (29.3)
Original article [39]	2017/2020	Egypt	Egypt (100)	Tilapia	100	29 (29.0)
Original article [40]	2019/2022	Vietnam	Unknown (103)	Undefined seafood	103	10 (9.7)
Original article [41]	2018/2019	Korea	Korea (120)	Clam	120	27 (22.5)
Original article [42]	2018/2020	Vietnam	Unknown (40)	Shrimp	40	24 (60.0)
Original article [30]	2012/2016	Vietnam	Vietnam (82)	Undefined seafood	82	37 (45.1)
Original article [43]	2015/2018	Algeria	Algeria (14)	Sardines	10	4 (40.0)
				Shrimp	4	0
Original article [44]	2017/2019	Portugal	Unknown (150)	Tuna (sushi)	30	1 (3.3)
				Seabass (sushi)	30	1 (3.3)
				Salmon (sushi)	30	1 (3.3)
				Snapper (sushi)	30	0
				Bramble shark (sushi)	30	0
Original article [45]	2019/2021	Egypt	Egypt (200)	Tilapia	100	14 (14)
				Mullet	100	5 (5.0)
Original article [46]	2019/2020	India	Unknown (79)	Piranha, Catfish, shrimp, carp, Snakehead, Eel, Puria, Barb, Potasi, Loach, perch, Bulla machi, Karkaria, featherback, Narva	79	65 (82.3)
Original article [47]	2016/2018	Spain	Unknown (97)	Undefined seafood	97	16 (16.5)
Original article [48]	2015/2016	Tanzania	Tanzania (196)	Tilapia	196	26 (13.3)
Original article [49]	2014/2015	Vietnam	Unknown (124)	Shrimp	60	45 (75.0)
				Undefined seafood	64	40 (62.5)
Original article [50]	2018/2020	China	China (411)	Snakehead	213	2 (1.0)
				Carp	198	1 (0.5)
Original article [51]	2016/2022	Tunisia	Tunisia (1716)	Sea bream	485	8 (1.7)
				Sea Bass	156	1 (0.7)
				Clam	1075	18 (1.7)
Original article [52]	2021/2023	India	India (17)	Undefined seafood	17	7 (41.2)
Original article [53]	2022/2023	Vietnam	Unknown (80)	Snakehead, tilapia, carp, catfish, anabas	80	3 (3.8)
Original article [54]	2019/2019	Thailand	Thailand (120)	Shrimp	120	N/A
Original article [55]	2004/2011	Korea	Unknown (N/A)	Undefined seafood	N/A	N/A

**Table 2 foods-12-03033-t002:** Methods used to isolate and characterize ESBL-producing bacteria in the 42 selected studies.

Reference	Bacterial Isolation	Detection of Presumptive ESBL	ESBL Confirmation and Typing
[15]	Sample homogenate enriched in peptone broth and plated onto Aeromonas base (RYAN) agar	Double-disk synergy test	PCR at class level
[16]	Sample homogenate enriched in peptone broth and plated onto Aeromonas base (RYAN) agar	Double-disk synergy test	PCR at class level
[17]	Sample homogenate enriched in peptone broth and plated onto thiosulphate citrate bile salts sucrose (TCBS) agar	Double-disk synergy test	PCR at class level
[18]	Sample homogenate enriched in peptone broth and plated onto tryptone bile X-glucuronide (TBX) agar containing 2 μg/mL of cefotaxime	Double-disk synergy test	PCR at group level
[19]	Sample homogenate enriched in peptone broth and plated onto thiosulphate citrate bile salts sucrose (TCBS) agar	Double-disk synergy test	PCR at class level
[20]	Sample homogenate enriched in peptone broth and plated onto chromogenic ECC (CHROMagar) agar containing 0.25 μg/mL meropenem	Double-disk synergy test	WGS
[21]	Sample homogenate enriched in peptone broth and plated onto violet-red bile glucose (VRBG) agar	Double-disk synergy test	PCR at class level
[22]	Sample homogenate enriched in tryptone broth and plated onto MacConkey agar	Double-disk synergy test	PCR at class level
[23]	Non-homogenized sample enriched in peptone broth and plated onto MacConkey agar containing 1 μg/mL cefotaxime	Double-disk synergy test	PCR at group level and WGS
[24]	Sample enriched in enterobacteria enrichment (EE) broth and plated onto MacConkey agar	Double-disk synergy test	PCR at class level
[25]	Sample homogenate enriched in peptone broth and plated onto thiosulphate citrate bile salts sucrose (TCBS) agar	Double-disk synergy test	PCR at class level
[26]	Sample homogenate enriched in tryptone soy broth (TSB) and plated onto xylose lysine deoxycholate (XLD) agar and hektoen enteric agar	Double-disk synergy test	PCR at class level
[27]	Sample homogenate enriched in EC broth and plated onto ESBL chromogenic (CHROMagar) agar	Double-disk synergy test	PCR at class level
[28]	Sample homogenate enriched in peptone broth and plated onto Aeromonas base (RYAN) agar	Double-disk synergy test	PCR at class level
[29]	Sample homogenate enriched in peptone broth and plated onto thiosulphate citrate bile salts sucrose (TCBS) agar	Double-disk synergy test	PCR at class level
[30]	Sample homogenate enriched in peptone broth and plated onto xylose lysine deoxycholate agar (XLD) and CHROMagar Salmonella (CHROMagar)	Double-disk synergy test	PCR at group level
[31]	Unspecified	MIC determination using broth microdilution	PCR at variant level
[32]	Unspecified	MIC determination using broth microdilution	PCR at variant level
[33]	Unspecified	MIC determination using broth microdilution	PCR at variant level
[34]	Unspecified	MIC determination using broth microdilution	WGS
[35]	Sample rinsed in MacConkey broth and rinse directly plated onto MacConkey agar containing 2 μg/mL cefotaxime	Double-disk synergy test	WGS
[36]	Sample homogenate enriched in peptone broth and plated onto xylose lysine deoxycholate (XLD) agar containing 2 μg/mL cefotaxime	Double-disk synergy test	PCR at variant level
[37]	Sample streaked onto thiosulphate citrate bile salts sucrose (TCBS) agar	Double-disk synergy test	PCR at class level
[38]	Sample homogenate enriched in peptone broth and plated onto chromogenic ECC (CHROMagar) agar	Double-disk synergy test	PCR at class level
[39]	Sample streaked onto Columbia Blood (CBA) Agar	Double-disk synergy test	PCR at variant level
[40]	Sample homogenate enriched in peptone broth and plated onto chromogenic ECC (CHROMagar) agar	Double-disk synergy test	PCR at group level
[41]	Sample homogenate enriched in peptone broth and plated onto chromogenic Rambach agar (CHROMagar)	Double-disk synergy test	PCR at class level
[42]	Sample homogenate enriched in peptone broth and plated onto thiosulphate citrate bile salts sucrose (TCBS) agar	Double-disk synergy test	PCR at group level
[30]	Sample homogenate enriched in peptone broth and plated onto chromogenic ECC (CHROMagar) agar containing 1 μg/mL cefotaxime	Double-disk synergy test	PCR at class level
[43]	Sample homogenate enriched in peptone broth and plated onto xylose lysine deoxycholate (XLD) agar	Double-disk synergy test	PCR at variant level
[44]	Sample homogenate enriched in peptone broth and plated onto Levine (EMB) agar with and without cefotaxime (2 μg/mL)	Double-disk synergy test	PCR at class level
[45]	Sample homogenate enriched in peptone broth and plated onto xylose lysine deoxycholate (XLD) agar	Double-disk synergy test	PCR at variant level
[46]	Sample homogenate enriched in Brilliant Green Bile Lactose (BGBLB) Broth and plated onto MacConkey agar containing 1 μg/mL cefotaxime	Double-disk synergy test	PCR at variant level
[47]	Sample homogenate enriched in peptone Broth and plated onto chromogenic ESBL (ChromID) agar containing 1 μg/mL cefotaxime	MIC determination using broth microdilution	PCR at variant level
[48]	Sample homogenized in 0.9% saline before plating onto MacConkey agar containing 2 μg/mL cefotaxime	Double-disk synergy test	WGS
[49]	Sample homogenate enriched in peptone broth and plated onto chromogenic ECC (CHROMagar) agar containing 1 μg/mL cefotaxime	Double-disk synergy test	PCR at group level
[50]	Sample homogenized in saline before plating onto chromogenic ESBL (ChromID) agar	Double-disk synergy test	PCR at group level
[51]	Sample enriched in peptone broth and plated onto MacConkey agar containing 2 μg/mL cefotaxime	Double-disk synergy test	WGS
[52]	Sample homogenate enriched in EC broth and plated onto sorbitol (MUG) agar with tellurite-cefixime supplement	Double-disk synergy test	PCR at class level
[53]	Sample homogenate enriched in peptone broth and plated onto CHROMagar Salmonella (CHROMagar)	Double-disk synergy test	PCR at class level
[54]	Sample streaked onto blood, MacConkey, and chocolate agar	Double-disk synergy test	PCR at class level
[55]	Unspecified	Double-disk synergy test	PCR at class level

**Table 3 foods-12-03033-t003:** Pooled proportions of ESBL types in ESBL-positive isolates from seafood belonging to different bacterial genera.

Bacteria isolated	No. ESBL-Positive Isolates	CTX-M Total	Undefined CTX-M Class	CTX-M-1 Group					CTX-M-9 Group				CTX-M Group 2	SHV	TEM
				Undefined CTX-M-1 Group	CTX-M-15	CTX-M-55	CTX-M-3	CTX-M-130	Undefined CTX-M-9 Group	CTX-M-14	CTX-M-27	CTX-M-32			
*E. coli*	521 (59.5)	481 (92.3)	174 (33.4)	79 (15.2)	81 (15.5)	11 (2.1)	0	0	126 (24.2)	6 (1.2)	3 (0.6)	1 (0.2)	0	10 (1.9)	36 (6.9)
*Salmonella* spp.	5 (4.2)	5 (100)	1 (20)	0	0	4 (80)	0	0	0	0	0	0	0	0	0
*S. enterica*	24 (16.7)	23 (95.8)	1 (4.2)	3 (12.5)	11 (45.8)	0	1 (4.2)	1 (4.2)	3 (12.5)	3 (12.5)	0	0	0	3 (12.5)	0
*Klebsiella* spp.	0	0	0	0	0	0	0	0	0	0	0	0	0	0	0
*K. pneumoniae*	57 (76)	50 (87.7)	19 (33.3)	1 (1.8)	26 (45.6)	0	0	0	3 (5.3)	1 (1.8)	0	0	0	9 (15.8)	4 (7)
*K. oxytoca*	41 (100)	35 (85.4)	35 (85.4)	0	0	0	0	0	0	0	0	0	0	6 (14.6)	13 (31.7)
*Enterobacter* spp.	5 (11.4)	4 (80)	3 (60)	0	0	0	0	0	0	0	1 (20)	0	0	1 (20)	0
*E. cloacae*	7 (31.8)	2 (28.6)	0	0	2 (28.6)	0	0	0	0	0	0	0	0	5 (71.4)	0
*Citrobacter* spp.	21 (48.8)	17 (81)	15 (71.4)	0	2 (9.5)	0	0	0	0	0	0	0	0	3 (14.3)	4 (19)
*Proteus* spp.	18 (64.3)	18 (100)	18 (100)	0	0	0	0	0	0	0	0	0	0	0	0
*Providencia* spp.	8 (38.1)	6 (75)	6 (75)	0	0	0	0	0	0	0	0	0	0	1 (12.5)	2 (25)
*Shigella sonnie*	2 (100)	2 (100)	2 (100)	0	0	0	0	0	0	0	0	0	0	0	0
Enterobacterales total	709 (49)	643 (90.7)	274 (38.6)	83 (11.7)	122 (17.2)	15 (2.1)	1 (0.1)	1 (0.1)	132 (18.6)	10 (1.4)	4 (0.6)	1 (0.1)	0	38 (5.4)	59 (8.3)
*Vibrio* spp.	151 (34.2)	151 (100)	106 (70.2)	18 (11.9)	0	0	0	0	27 (17.9)	0	0	0	0	0	0
*Aeromonas* spp.	26 (25.2)	26 (100)	26 (100)	0	0	0	0	0	0	0	0	0	0	0	0
*Other* spp. ^a^	34 (56.7)	27 (79.4)	8 (23.5)	13 (38.2)	0	0	3 (8.8)	0	0	0	0	0	3 (8.8)	7 (20.6)	1 (2.9)
Total Isolates	920 (44.8)	847 (92.1)	414 (45)	114 (12.4)	122 (13.3)	15 (1.6)	4 (0.4)	1 (0.1)	159 (17.3)	10 (1.1)	4 (0.4)	1 (0.1)	3 (0.3)	45 (4.9)	60 (6.5)

^a^ Includes species of genera: Serratia, Acinetobacter, Pseudomonas, Edwardsiella, Erwinia, Hafnia, and Yersinia.

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
