# Peer review of "Prevalence and Types of Extended-Spectrum β-Lactamase-Producing Bacteria in Retail Seafood"

_foods, 2023, doi:10.3390/foods12163033_

Round 1

Reviewer 1 Report

The review manuscript presented by Pearce and Guardabassi gathers the most relevant information regarding epidemiology of the ESBL genes in Gram-negatives present in fish and seafood worldwide.

Overall, the study is well conducted but I have serious concerns in terms of publication in its current form, due to the (very) limited amount of studies that were included in the meta-analysis. Even though for a mini-review the amount of manuscript reviewed is rather short, making the general vision of the state-of-the art too narrow, from my point of view. I would reconsider the whole manuscript including more papers published to give the readers a broader perspective on the topic.

Suggestions intended to improve the quality of the manuscript

-          To avoid misunderstanding, I would modify the title to “fish and seafood” rather than only seafood.

-          In sections 2.2. and 2.3., please include a flow diagram to explicitly display the steps followed in the meta-analysis and the number of documents retrieved in each one of the steps.

-          Lines 127-129: All these names must be in italics. Please correct throughout the manuscript.

-          It is no my intention to undermine the work performed by the authors, but without any statistical treatment, the manuscript turns out to be merely descriptive and an enumeration of results published elsewhere. For example, is there any relationship between the product and the genes present? Or is there any statistical evidence that in a given country, the incidence/distribution of a specific ESBL is significant?

Reviewer 2 Report

Dear author

I would like to thank you for the interresting review I had been able to review. I think it will be an interesting contribution to the litterature. A lot of studies are starting on seafood and AMR you are right in time.

1/let's start with the typos/lettering/small details...

lines 41-42: the  names of the beta-lactamase genes are not spelled properly : blaTEM

line 173 : do you mean blaSHV-27 ? please amend 

line121 : from the table1 I have guessed "Algeria (n=1)" Am I right ?

line 126-129, line 165, 169(x2), line 206 : names of the bacteria should be in italics ?

line 254 "LMIC" please avoid unnecessary abbreviations. Spell it out, it would be easier for the readers to follow your ideas.

line 276-281: authors contributions should be cleaned from line 276-277 and double checked

reference mistake  : numbers are appearing twice. some useless hyphen are everywhere. Please clean this section which is specially important for a review article.

2/ Materials and methods is quite good but can still be slightly improved.

I like your reference #10 in PROSPERO. I found it easily and it replied pretty well to my questions on section 2.2 about the detail search. For the naïve readers like me, it would be nice to state in the section 2.2 that the advance search criteria are details in prospero registration.

I would also introduce here a small section 2.7 called limitations where I would copy and paste ( and improve if needed) the section from end of discussion about "systematic reviews" line 255-259. I think it fits better here. Do you agree ?

3/Tables : very informative 

table1 ok

table2 would you be kind enough to organise by groups the CTX-M. It would ease reading ( CTX-M-15 column shouldn't be fare from the column about CTX-M-1 group. a bold line when you switch to SHV or TEM would also be nice to read

table3 : please be careful with the taxonomy! Proteus and Providencia are members of the Morganellaceae family. So please move them out of the enterobacteriaceae family. You have also to get them out of the total. Table 3 needs to be reworked

4/ discussion : very interesting but need sometimes to be more precise.

line217-219 : reference #57 does notapply very well to this sentence. If I am wrong, please prove me I am wrong and accept my appologies. otherwise find some new references to prove that CTX-M is the most pervasive ...

line241 : don't limit your sentence to japanneese recepies, you seem to forget about the consumption of raw molluscs in developped countries, .. you should get figures about seafood consumed raw more precise than just feelings.

line 246-247 : refrence is absolutely needed ? FAO figures ? data are clearly missing here to support this very interesting idea.

line 259-264 : this idea of the limitations you identify in your review is very interesting and of much value to your audence. I am frustrated, I would like you to elaborate a much more detailed paragraph. It is too fast.

5/ conclusions : not as good as other parts of the article

sentence 266-269 is not in line with what you stated before in the introduction around lines 40 to 50. I am very confused by this sentence. Isolating ESBl from seafood does not prove that's the way humans are contaminated, does it ?

please write a conclusion in line with the entire article.
